# Late-life fitness gains and reproductive death in *Cardiocondyla obscurior* ants

**Luisa Maria Jaimes-Nino\*, Jürgen Heinze, Jan Oettler\***

Zoologie/Evolutionsbiologie, Universität Regensburg, Regensburg, Germany

**Abstract** A key hypothesis for the occurrence of senescence is the decrease in selection strength due to the decrease in the proportion of newborns from parents attaining an advanced age – the so-called selection shadow. Strikingly, queens of social insects have long lifespans and reproductive senescence seems to be negligible. By lifelong tracking of 99 *Cardiocondyla obscurior* (Formicidae: Myrmicinae) ant colonies, we find that queens shift to the production of sexuals in late life regardless of their absolute lifespan or the number of workers present. Furthermore, RNAseq analyses of old queens past their peak of reproductive performance showed the development of massive pathology while queens were still fertile, leading to rapid death. We conclude that the evolution of superorganismality is accompanied by 'continuusparity,' a life history strategy that is distinct from other iteroparous and semelparous strategies across the tree of life, in that it combines continuous reproduction with a fitness peak late in life.

## Editor's evaluation

The article will contribute significantly to our understanding of superorganism development and longevity in ants and will provide testable hypotheses in other ant species and other organisms.

**\*For correspondence:**
jaimes.luisa@outlook.com
(LMJ-N);
joettler@gmail.com (JO)

**Competing interest:** The authors declare that no competing interests exist.

## Introduction

The phenomenon that social insect queens live exceptionally long compared to solitary insects is widely recognized (*Keller and Genoud, 1997*; *Carey, 2001*). Given how prominent this is, however, only patchy information exists about the proximate mechanisms that are involved with the regulation of senescence, that is, a phase marked by an increase in relative mortality and a decrease in relative fecundity with age. Even less is known about the ultimate causes of social insect aging.

The classic trade-off between reproduction and maintenance shapes the life history strategy of species along a continuum between iteroparity (repeated events of reproduction) and semelparity (single event of reproduction) (*Hughes, 2017*). These strategies shape the way species age, that is, how resources are allocated to maximize fitness. In iteroparous species, fitness decreases after the first reproductive peak(s). Thus, the strength of selection against age-specific mortality decreases with age as the proportion of offspring that come from parents surviving to a specific age becomes smaller with time (*Hamilton, 1966*; *Moorad et al., 2020*). This is known as the selection shadow, which begins with maturity (*Williams, 1957*) and may negatively affect reproductive performance and survival (i.e., senescence). Classic model systems in aging research, such as *Drosophila*, *Caenorhabditis* (but see the discussion of quasi-semelparous hermaphrodites, *Gems et al., 2021*), mice, and humans, are of the iteroparous type, and a plethora of studies have revealed common mechanisms associated with senescence rate (*Gems and Partridge, 2013*). One prominent evolutionary theory of aging explains senescence by genes with antagonistic pleiotropic effects early and late in life (*Williams, 1957*). Semelparity instead predicts that organisms optimize their resources to one fitness peak, after which reproductive death occurs, that is, allocation of remaining resources into fecundity and not into maintenance. Thus, selection acts strongly against senescence before the single reproductive event. To

understand how investment in reproduction of ant queens changes with chronological age and how social insect queens age, it is vital to investigate where they sit on this parity continuum.

Two dimensions are necessary to understand aging from an evolutionary perspective: the pace and the shape of demographic trajectories (*Baudisch, 2011*; *Baudisch et al., 2019*). The pace refers to factors that describe the timescale (e.g., life expectancy), and the shape refers to time-standardized measures of the distribution of mortality and fertility across a life history. Studies that capture the shape of aging (age-specific reproduction and mortality) of ants are scarce (reviewed in *Cole, 2009*), and based on punctual periods of growth and death of colonies, mostly of long-lived species in which colonies have a single queen (monogyny). Often such field data correspond to less than 20% of the estimated lifespan of the species (*Atta cephalotes*, *Perfecto and Vandermeer, 1993*; *Atta colombica*, *Wirth et al., 2003*; *Pogonomyrmex owyheei*, *Porter and Jorgensen, 1988*; *Pogonomyrmex occidentalis*, *Keeler, 1993*; *Pogonomyrmex badius*, *Tschinkel, 2017*). These studies have generally failed to capture the end of the queen's lifespan and thus did not document senescence and lifetime reproductive investment. More complete yearly census data from *Pogonomyrmex barbatus* showed no relation between reproductive success (number of successfully established offspring colonies) and age (*Ingram et al., 2013*), but an increase in the production of male and female sexuals with age (*Wagner and Gordon, 1999*). In contrast, a study on a related species, *P. occidentalis*, showed no correlation between sexual production and colony size (as a proxy for age) once colonies had initiated sexual reproduction (*Cole and Wiernasz, 2000*), suggesting that it is difficult to infer the dynamics of age, colony growth, and reproduction from field data. To better understand how aging, senescence, and reproductive investment are related in ants, complete lifetime production data of individual queens are needed.

To study aging patterns and senescence of social insect queens, it is helpful to consider the colony as a superorganism (*Wheeler, 1911*; *Boomsma and Gawne, 2018*), analogous to a soma- (i.e., workers) and a germline (i.e., queens), where the investment into both castes is related and affects overall fitness (*Bourke, 2007*; *Kramer and Schaible, 2013*). Ant species such as *Cardiocondyla obscurior*, in which workers are completely sterile and seemingly without any direct reproductive power, exhibit an extreme case of superorganismality. By manipulating colony size, we expected to find trade-offs between lifespan and investment in queen/worker/male offspring. We monitored the lifetime production of individual queens in 99 single-queen colonies maintained with 10, 20, or 30 workers each (*Figure 1—figure supplement 1A and B*). Worker number corresponds to the colony size variation observed in the field (*Schrader et al., 2014*, *Figure 1—figure supplement 2*) and was standardized weekly. Queens whose egg production declined below a rate of ~10 eggs/week exhibited lethargic behavior, were less mobile, left the nest and/or were harassed by workers, and died within a few days to weeks. To assess if senescence was restricted particularly to the end of life, we compared RNAseq data of 18 of such *prope mortem* (Lat. near death) queens (between 28 and 49 weeks old) and 18 middle-aged queens (between 19 and 21 weeks), which were in their peak of fertility (*Figure 1—figure supplement 3A and B*). To compare queen and worker mortality, we tracked the survival of 40 workers kept in 40 colonies with either 10 or 20 nestmate workers.

## Results

### Reproductive strategy

The treatment (varying worker number) did not affect total production of eggs (package 'generalized linear mixed models using template finder' v. 1.1.2.3 in R) (*Figure 1A*, 10 vs. 20 workers: glmmTMB z-value = –0.38, p=0.70 and 10 vs. 30: z-value = –0.96, p=0.34) or worker pupae (*Figure 1B*, 10 vs. 20 workers: glmmTMB z-value = 0.09, p=0.93 and 10 vs. 30: z-value = –0.39, p=0.70). The treatment also did not affect the lifespan of queens (*Figure 1—figure supplement 6A*, Cox proportional hazard regression model, likelihood ratio test, $X^2$ = 1.57, p=0.46), which was highly variable across treatments (variation coefficient: 32.2%, *Figure 1—figure supplement 6B*).

We hypothesized that colonies that experienced a worker shortage would compensate by investing less into the production of new queens as these are larger and therefore more costly to produce. Indeed, queens with 10 workers (n = 31) produced significantly fewer queen pupae than queens with 20 (n = 34) (glmmTMB z-value = 2.81, effect size=1.97, p=0.005) and 30 workers (n = 34) (glmmTMB z-value = 2.58, effect size=1.78, p=0.009, *Figure 1C*) with no significant differences between 20 and

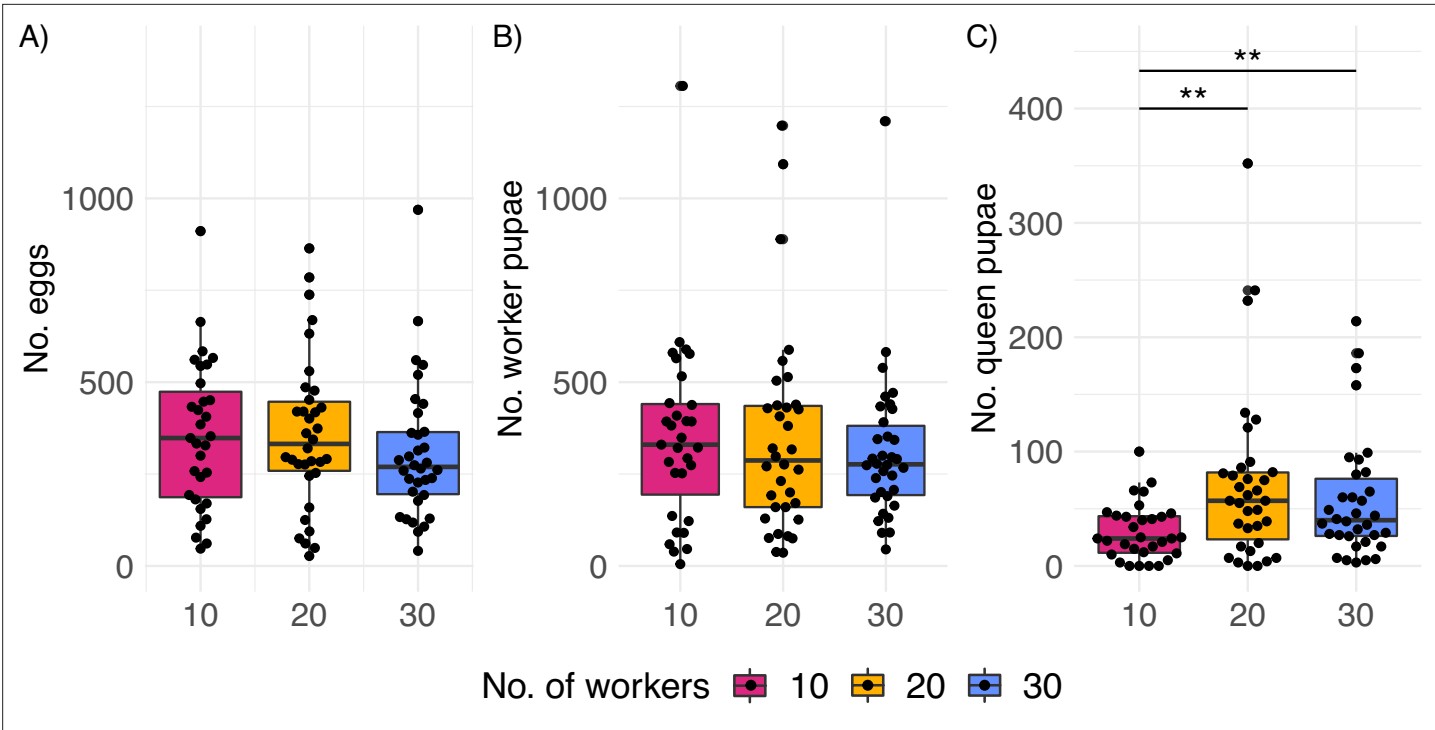

**Figure 1.** Productivity of *C. obscurior* colonies across treatments. (**A**) Total number of eggs, (**B**) worker pupae, and (**C**) queen pupae (N = 31, 34, and 34 for 10, 20, and 30 worker colonies, respectively). Significant differences are given with **p<0.01 and ***p<0.001. Boxplots depict upper and lower quartile plus 1.5 interquartile range (IQR).

The online version of this article includes the following figure supplement(s) for figure 1:

**Figure supplement 1.** Number of workers in the experimental set-up.

**Figure supplement 2.** Worker numbers of colonies collected in 2013 (N = 27) and 2018 (N = 52) in Brazil and in 2011 in Japan (N = 62).

**Figure supplement 3.** Productivity at the time of sampling of *prope mortem* (28–49 weeks old) and middle-aged queens (19–21 weeks old), n = 18.

**Figure supplement 4.** Egg productivity.

**Figure supplement 5.** A queen of *C. obscurior*.

**Figure supplement 6.** Queen longevity.

**Figure supplement 7.** Investment caste ratio as total investment into queens over total investment into queens and workers (as Queen*c/[Queen*c + Worker]).

**Figure supplement 8.** Wingless males, so-called ergatoid males, produced per week in *C. obscurior* colonies across treatments.

30 workers (glmmTMB z-value = –0.49, p=0.877). Similar results were obtained when accounting for the differences in biomass between workers and queens (*Figure 1—figure supplement 7*, *Supplementary file 1A*). Probably due to difficulties assessing precise egg numbers which are reared in piles, and extremely worker-biased caste ratios (average pupae developed into workers = 0.86), egg counts do not reflect these subtle but significant differences. The median sex ratio (queen/queen + male pupae) across treatments was 0.85 (25 and 75% quantiles = 0.79 and 0.90), and total production of male pupae (two types of males occur in *C. obscurior*: winged and wingless) was unaffected by the treatment (10 vs. 20: glmmTMB z-value = 1.94, p=0.05 and 10 vs. 30: glmmTMB z-value = 1.52, p=0.13, *Figure 1—figure supplement 8*). Queens produced very low numbers of winged males during their lifetime (mean = 0.36, median = 0, N = 99).

A first peak in the investment in queen pupae occurred around 15 weeks after the colonies were established (*Figure 2A*), followed by an increasing queen bias with age (*Figure 2B*). In general, new queens, which start a new colony, invest first in growing numbers of workers (ergonomic phase) and subsequently in the production of new sexuals, when the colony has reached the threshold required to enter the reproductive phase (*Macevicz and Oster, 1976*; *Oster and Wilson, 1978*; *Beekman et al., 1998*). This shift in caste ratio does not result from a drop in the production of pupae at the

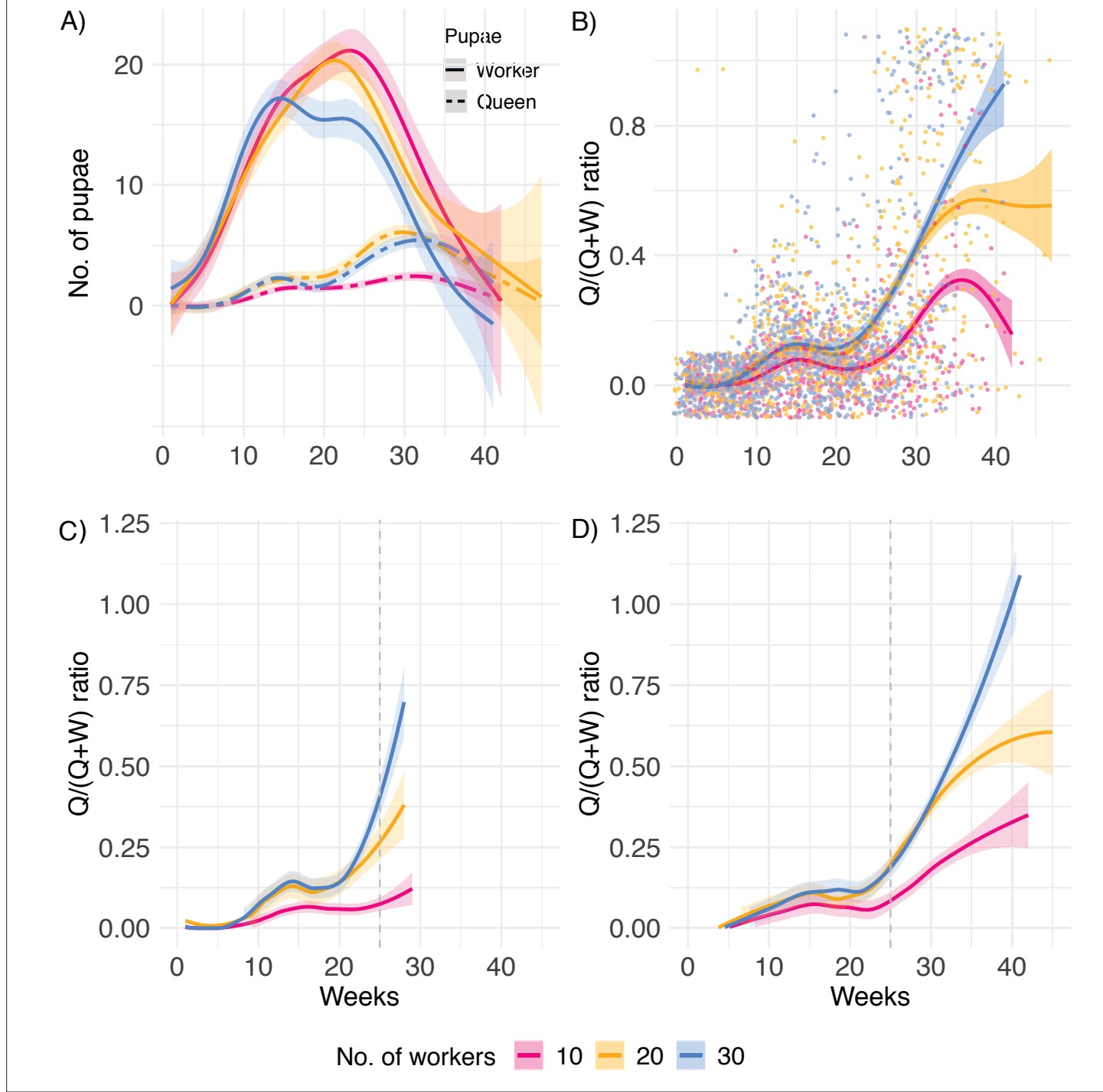

**Figure 2.** Lifetime investment. (**A**) Numbers of worker and queen pupae produced over time, (**B**) queen/(queen + worker) pupae caste ratio produced by queens (n = 31, 34, and 34 for 10, 20, and 30 worker colonies, respectively), (**C**) pupae caste ratio for queens with lifespan below (n = 44), and (**D**) above the mean lifespan of 25 weeks, indicated by the dashed line (n = 55). After the queen's death, eggs and larvae were allowed to develop into pupae for a final count. Therefore, smooth splines extend ca. 4 weeks after queen death.

The online version of this article includes the following figure supplement(s) for figure 2:

**Figure supplement 1.** Production of pupae and eggs per queen over time before death (n = 99).

**Figure supplement 2.** Correlation between counted eggs and developed adults.

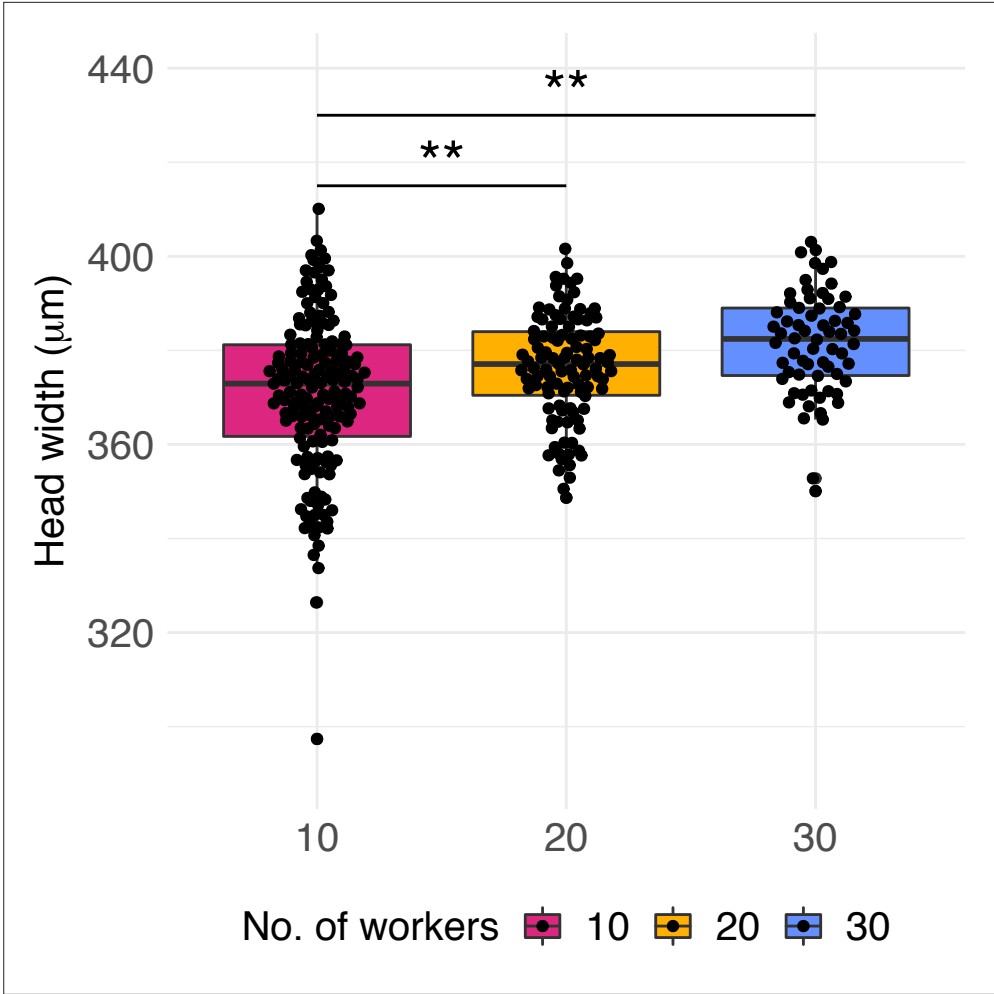

**Figure 3.** Worker quality across treatments. Head width measurements of workers produced by queens of colonies with 10, 20, and 30 workers (n = 160, 112, and 68, respectively). Significant differences are given with * p<0.05 and **p<0.01. Boxplots depict upper and lower quartile plus 1.5 interquartile range (IQR).

end of life. In contrast, pupa production is at its highest just before death (*Figure 2—figure supplement 1*). Importantly, in *C. obscurior* this caste ratio shift appeared to be a fixed trait, independent of colony size and queen lifespan. Both queens with short and long lifespans (below and above the mean lifespan of 25 weeks, *Figure 2C and D*, respectively), equivalent to queens with low and high productivity, exhibited late-life investment into queens.

In addition to the effect on caste ratio, the treatment had an effect at the colony level. We explored whether the quality of workers was affected by measuring the head width of workers produced over months 3–6 of the queen's lifetime (approximately five workers per month). Head width of workers was 2% and 3% significantly smaller in colonies with 10 workers than in colonies with 20 (glmmTMB z-value = 2.22, p=0.026) and 30 workers, respectively (glmmTMB z-value = 2.68, p=0.007, *Figure 3*), but not different between colonies with 20 and 30 workers (glmmTMB z-value = 0.22, p=0.97). This suggests that small colonies lack sufficient numbers of nurse or forager workers, and indeed colonies collected in the field have worker numbers closer to 20 or 30 (*Schrader et al., 2014*; *Figure 1—figure supplement 2*).

After mean-standardizing queen age-specific mortality and fecundity (following *Jones et al., 2014*), we found that relative fecundity reached its maximum after ~16 weeks, before completion of the median lifespan (~25 weeks), and then decreased (*Figure 4*). Production of workers tightly followed the curve of egg production. Importantly, relative investment in queen and male pupae reached its maximum late in life (~28 weeks). This pattern is not due to the delay in development

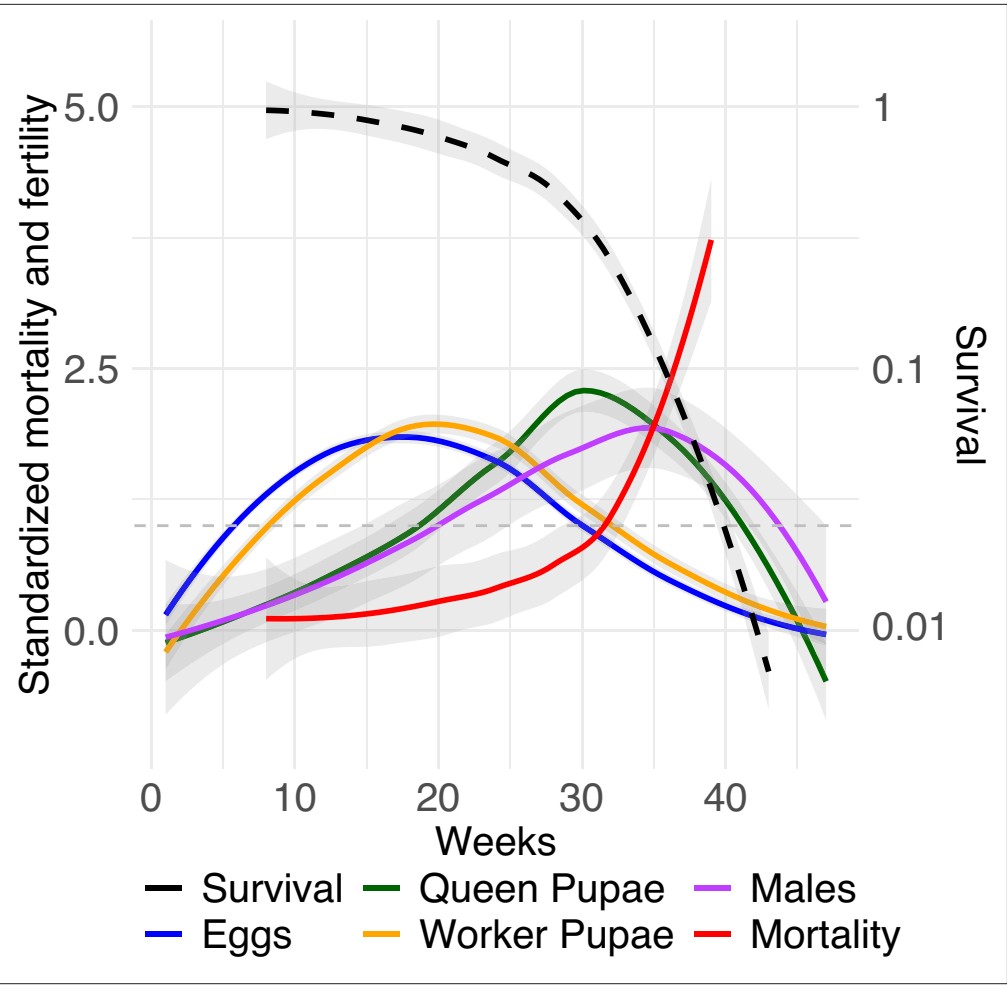

**Figure 4.** Relative mortality and fecundity as a function of age. Mean standardization of age by dividing age-specific mortality and fecundity of queens (n = 99) by their means after maturation (*Jones et al., 2014*). Survivorship (black dashed line) is depicted on a log scale. The graph uses a Loess smoothing method (span = 0.75) and a confidence interval of 95%. The dashed gray line at y = 1 indicates when relative mortality and fertility are equivalent to mean mortality and fertility.

The online version of this article includes the following figure supplement(s) for figure 4:

**Figure supplement 1.** Age-specific mortality of ant queens (n = 135).

**Figure supplement 2.** Relative mortality of worker as a function of age.

from egg to pupa because queen and male development only lasts ~5 and ~ 3 weeks, respectively (*Schrempf and Heinze, 2006*). Furthermore, *C. obscurior* ant queens exhibited a below-average level of adult mortality until week 30, after which mortality increased above the average level (*Figure 4*, *Figure 4—figure supplement 1*). This indicates maintenance of selection until after the peak of relative investment in sexual offspring. Therefore, queens continue to experience strong selection even at high ages, that is, weeks after they reached the mean lifespan. Monitored workers in colonies with 10 or 20 nestmates did not differ in survival (Cox proportional hazard regression model, likelihood ratio test, $X^2$ = 0.06, p=0.8). Therefore, the mean-standardized age-specific mortality was calculated for the 40 workers. Note that regardless of the differences in timescale, the shape of mean-standardized mortality of workers was similar to that of queens (*Figure 4—figure supplement 2*). This suggests that aging is a genetically fixed trait expressed by queens and workers alike.

## RNAseq of *prope mortem* queens

To determine if queens show signs of reproductive senescence and loss of physiological function, we analyzed gene expression data of *prope mortem* queens exhibiting decreasing egg-laying rates

and middle-aged queens that were at their peak reproductive performance. To account for possible effects of fertility, we sampled queens with low, medium, and high egg productivity at 18 weeks of age (*Figure 1—figure supplement 4A and B*). We subjected the head plus thorax and the gaster (see methods for terminology, *Figure 1—figure supplement 5*) to RNAseq separately to assess if reproductive tissue shows a different physiological wear and tear than head-thorax tissue. The analyses revealed that head-thorax and gaster tissues showed similar mapping rates to the genome (*Figure 5—figure supplement 1A and B*), but that gaster samples had a lower GC content on average and more duplicated reads (*Figure 5—figure supplement 1C–F*) in *prope mortem* queens compared to middle-aged queens.

Of the 20,006 expressed genes in head-thorax tissue, 3565 (17.8%) genes were differentially expressed between middle-aged and *prope mortem* queens (after false discovery rate [FDR] adjustment p<0.001, DESeq2, *Source data 9*). Of these, 1725 genes (48%) were upregulated and 1840 genes (52%) were downregulated in *prope mortem* queens compared to middle-aged ones. Gene Ontology (GO) term enrichment revealed signs of rapid physiological decay of *prope mortem* queens, such as reduced translation, proteasomal, ribosomal, and mitochondrial function (Fisher test using the weight01 algorithm, p<0.05, *Supplementary file 1B*, *Figure 5*), increased splicing, and transcript processing (*Supplementary file 1C*, *Figure 5—figure supplement 2*). Such processes have previously been related to aging in several model organisms (*López-Otín et al., 2013*); for example, the loss of protein homeostasis (*Hipp et al., 2019*), the decrease in ribosomal proteins (*Walther et al., 2015*), alterations in the mitochondrial function (*Green et al., 2011*), disruption of splicing (*Bhadra et al., 2020*), and others (*Harries et al., 2011*). Another characteristic of aging, changes in gene connectivity among gene expression networks found in mice (*Southworth et al., 2009*), was not affected by age in *C. obscurior* (calculated using the softConnectivity and the biweight midcorrelation functions on gene networks for middle and *prope mortem* queens using WGCNA, and modeled using glmmTMB, z-value = −1.7, p=0.09). Principal component analysis (PCA) ordination of the head-thorax tissue separated middle-aged and *prope mortem* queens by age (PERMANOVA test, F-value = 7.59, p<0.001), but not by fertility (F-value = 1.09, p=0.26) or duplication percentage (*Figure 5—figure supplement 3A and B*).

In the gaster tissue, 4832 (24.3%) of 19,925 expressed genes were differentially expressed between age groups (after FDR adjustment p<0.001, DESeq2, *Source data 10*). Of these, 2306 genes were upregulated (48%) and 2526 downregulated (52%) in *prope mortem* queens compared to middle-aged queens. GO term enrichment likewise showed that many fundamental processes were affected in *prope mortem* queens, such as DNA damage, telomere maintenance, and enrichment of transcription processes (*Supplementary file 1D*, *Figure 5—figure supplement 4*), and among others processes related to protein processing, glycolytic processes, and the Notch signaling pathway were downregulated (*Supplementary file 1E*, *Figure 5—figure supplement 5*). In contrast to head-thorax tissue, gene connectivity among gene co-expression networks in the gaster was significantly different (*prope mortem* queens: median = 69.45; middle-aged queens: median = 52.56) (glmmTMB, z-value = −19.5, p<2e-16), but contrary to what was found for aged mice (*Southworth et al., 2009*).

The PCA of the 500 most variable expressed genes in gaster tissue shows that the samples group according to age (PERMANOVA test, F-value = 13.91, p<0.001), fertility level (F-value = 1.95, p=0.04), and also the percentage of duplicated reads in the libraries (F-value = 4.83, p=0.002, *Figure 5—figure supplement 6A and B*). This is not a typical technical artifact (no correlation to sequencing lane, RNA concentration, or quality). Spike-in reads were used as a control for library preparation and showed a positive linear relationship between expected and observed reads independent of age group, tissues, and lanes (*Figure 5—figure supplement 7A–C*). However, this linear relationship has different slopes among age groups in the gaster samples (*Figure 5—figure supplement 7B*), indicating biological changes with age pertaining specifically to the gaster.

In spite of these discrepancies between tissue types, 104 GO terms were significantly enriched in both tissues, of these 44 in *prope mortem* queens (*Supplementary file 1F*) and 60 in middle-aged queens (*Supplementary file 1G*). Thus, signs of similar physiological pathologies occur in reproductive and non-reproductive tissue.

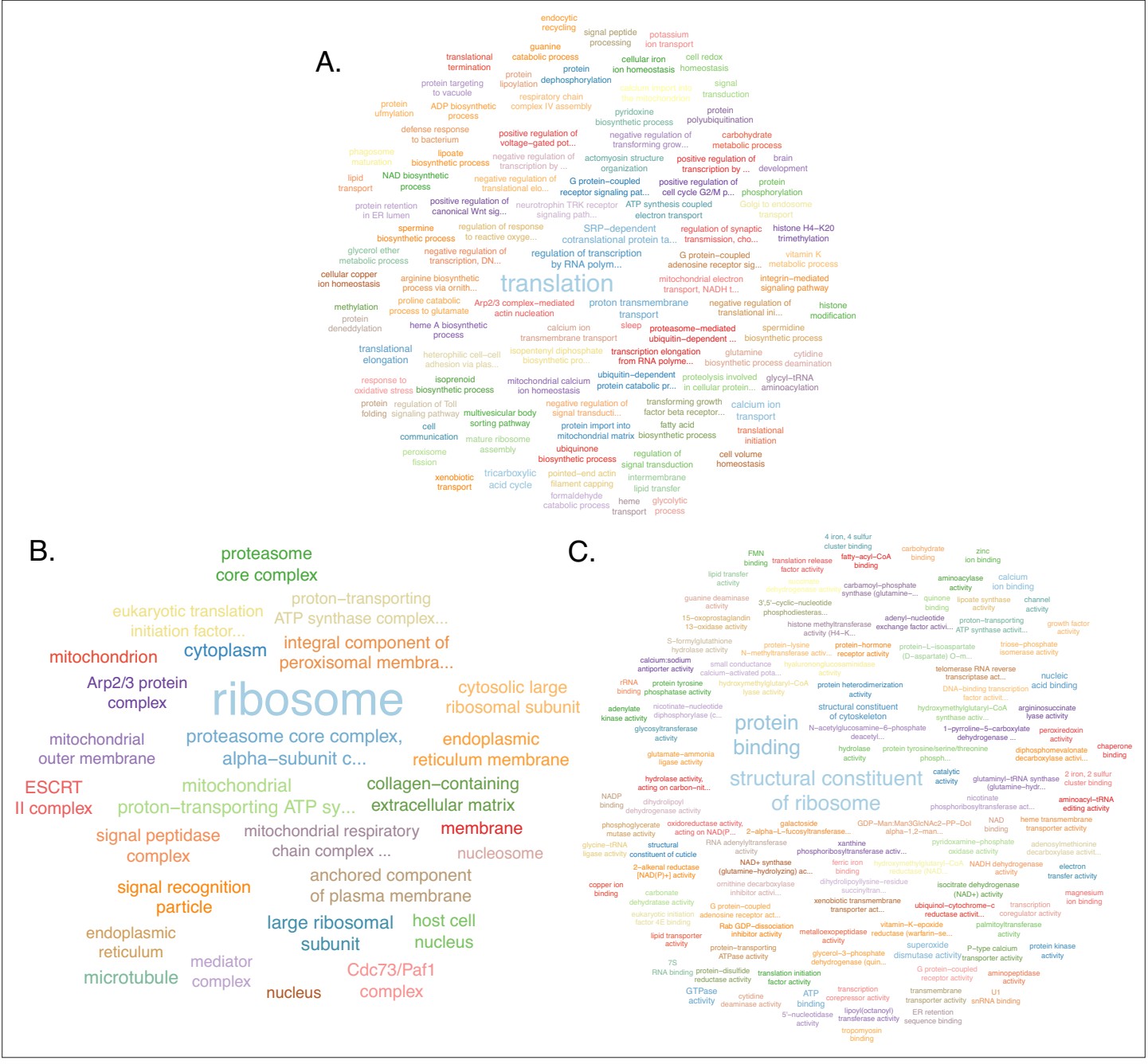

**Figure 5.** Enriched Gene Ontology (GO) terms downregulated in *prope mortem* queens compared to middle-aged queens in the head-thorax tissue. Functional annotation and enrichment analysis using topGO (version 2.46.0) and the weight01 algorithm to calculate significance for (**A**) biological processes, (**B**) cellular components, and (**C**) molecular functions.

The online version of this article includes the following figure supplement(s) for figure 5:

**Figure supplement 1.** Mapping and GC content percentage.

**Figure supplement 2.** Enriched Gene Ontology (GO) terms upregulated in *prope mortem* queens compared with middle-aged queens in head-thorax tissue.

**Figure supplement 3.** Principal component analysis (PCA) 1 and 2 of head-thorax samples using the top 500 most variable expressed genes after variance-stabilizing transformation.

**Figure supplement 4.** Enriched Gene Ontology (GO) terms upregulated in *prope mortem* queens compared with middle-aged queens in the gaster.

**Figure supplement 5.** Enriched Gene Ontology (GO) terms downregulated in *prope mortem* queens compared with middle-aged queens in the gaster.

*Figure 5 continued on next page*

*Figure 5 continued*

**Figure supplement 6.** Principal component analysis (PCA) 1 and 2 of gaster samples using the top 500 most variable expressed genes after variance-stabilizing transformation.

**Figure supplement 7.** Sequencing quality check based on Spike-in reads.

## Discussion

The near-ubiquitous occurrence of senescence has been explained by two classic prevailing evolutionary theories, mutation accumulation and antagonistic pleiotropy (Medawar, 1941; *Williams, 1957*). These theories have in common the basic assumption of the existence of a 'selection shadow': a decrease in the force of natural selection with age. The selection shadow leads to loss of function and senescence, that is, an increase in relative mortality and a decrease in relative fecundity with age (*Maklakov and Chapman, 2019*). Originally explained as a consequence of extrinsic mortality, models have shown that the strength of selection is in fact influenced by the proportion of offspring coming from parents that survived to a certain age (*Hamilton, 1966*; *Moorad et al., 2020*). Extensive demographic data show a huge diversity of aging patterns across metazoan species, ranging from 20 times the average mortality at terminal age to less than a half in other species (*Jones et al., 2014*; *Cohen, 2018*). In some cases, a short phase of senescence is self-evident, for example, in semelparous species such as salmon, where death follows reproduction to provision the next generation with resources.

Here, we show for the first time the shape of aging in a social insect. While fecundity decreases in the ant *C. obscurior*, reflecting reproductive senescence, investment into sexuals reaches a maximum late in life, regardless of individual fitness (queen lifespan or total egg productivity) and colony size. Males in this species usually transfer an excess amount of sperm (*Schrempf and Heinze, 2008*), and only one queen showed signs of sperm depletion and produced only males at the end of her life. Therefore, reproductive senescence cannot be explained by sperm depletion. The magnitude of the investment (i.e., number of queen pupae produced) is affected by the number of workers available. In *C. obscurior*, most new queens were produced by queens older than the mean queen lifespan, indicating that queens continue to experience strong selection at high ages. This is in line with the hypothesis that the strength of selection against age-specific mortality is proportional to the probability for any offspring in the population to be produced by parents of that age and older.

Strikingly, relative mortality did not increase directly after maturity or after total egg production started decreasing, but after the production of sexual pupae had reached its maximum. IAccordingly, transcriptome profiles of *prope mortem* queens shortly after this investment peak, which produced decreasing numbers of eggs, revealed signs of a broad range of physiological pathologies. The changes seemed stronger in the gaster (e.g., percentage of duplicated reads), which contains the reproductive organs and most of the digestive system, but to a similar extent occurred in head and thorax, containing most neuronal and muscle tissue. Such a systemic breakdown is expected assuming that the entire physiology is optimized towards a fitness peak. Strikingly, a comparative transcriptomic study of 8-week young queens with fully mature *C. obscurior* queens at or close to their peak fecundity (18 weeks old) did not find signs indicative of aging, but in comparison to aged *Drosophila* flies an opposite regulation of processes (e.g., cellular ketone, carbohydrate, and organic acid metabolic processes) and genes (e.g., *ref(2)P*, *emp*, *P5cr-2*, *CCHa2*, *NLaz*, *Sirt6*) involved in aging (*von Wyschetzki et al., 2015*). Furthermore, a gene co-expression network study using the same data showed higher connectivity in middle-aged queens, indicating increased transcriptional regulation with age (*Harrison et al., 2021*). Together, this suggests that the physiology of queens is maintained until the fitness peak is reached, at which time they undergo physiological deterioration, while still being reproductively active. This pattern is reminiscent of semelparous species with reproductive death rather than that typical of iteroparous species in which selection against age-specific mortality decreases after a first reproduction event and actuarial senescence unfolds under the selection shadow.

## Conclusion

Superorganismality is a major evolutionary transition, and this transition is accompanied by a change in the mode of reproduction. We propose that the evolution of 'continuusparity' (Lat.: 'continuus' meaning incessant/successive; and 'parere' meaning giving birth), that is, the combination of lifelong

continuous reproduction and increasing fitness returns late in life, underlies the delay of the selection shadow, the maintenance of selection strength against age-specific mortality, a brief phase of senescence late in life, and finally reproductive death. This is not to be confused with the meaning of the term negligible senescence as actuarial and reproductive senescence clearly occur at the end of life.

'Continuusparity' emerges as a combination of iteroparous and semelparous characteristics: reproduction resembling continuous iteroparous species but without the inter-parous nonreproductive breaks, during which nests are built, mating occurs, resources are acquired, etc. The iteroparous solitary ancestor of ants is thought to be related to mud dauber wasps (Sphecidae) and cockroach wasps (Ampulicidae) (*Ward, 2014*), parasitoid wasps with mass provisioning. This combines with aging/resource allocation patterns of semelparous species (which are in contrast mostly short-lived), optimized towards one reproductive episode at the end of life, followed by reproductive death. Continuous reproduction is possible because no extra time or energy is necessary for further acquisition of resources, brood care, territoriality, etc., because of the extended phenotype, the colony. With time the colony increases in size, and resources increase accordingly, analogous to solitary organisms with lifelong growth (*Keller, 1998*). Continuous reproduction is further facilitated by the presence of a spermatheca in female insects, which allows for a single mating event and lifelong sperm storage. Thus, the costs of additional matings are zero.

We propose that continuusparity and its effect on the shape of queen aging is a property of superorganismality, and that this life history strategy ultimately underlies the evolution of long lifespans in social insects. For the pace of aging, it is important whether queen and worker interests are aligned, and whether direct and indirect reproductive investments of queens and workers are optimized. With respect to which proximate mechanisms regulate aging in social insects, this framework predicts that there are no genes/pathways with antagonistic pleiotropic effects because there is no 'later in life' past the fitness peak. Under this perspective, many questions remain open. Queens appear to have different properties that influence both lifespan and fertility (*Kramer et al., 2015*), probably determined during larval development (*Schultner et al., 2017*), and which are key to understanding why some individuals live longer than others. What are these properties, and how are they determined, maintained, and finally terminated? Queens do not senesce until shortly before death, so how is the trade-off between somatic maintenance and reproduction resolved? Using this framework, we can now start to study the proximate regulators that maintain the homeostasis in *C. obscurior* ant queens, which remain hidden in the excess number of associated processes.

## Materials and methods
### The species
*C. obscurior* is probably the best studied ant species with respect to aging due to the relatively short lifespan of queens (~6 months). Colonies comprise a few queens (body length ~3 mm), a few dozen workers (~2 mm), and nest in small cavities in dead twigs, aborted fruits, rolled leaves, under bark, etc., in trees and shrubs (*Schrader et al., 2014*). Virgin queens usually mate once with related wingless males inside the natal colony (*Heinze and Hölldobler, 1993*; *Schmidt et al., 2016*; *Heinze and Hölldobler, 2019*), generally stay in the nest, and new colonies are formed by budding of colony fragments. This mode of reproduction from small propagules allows for successful colonization of disturbed habitat in warm climates around the world (*Heinze and Delabie, 2005*; *Heinze, 2017*). Various social, environmental, and biotic factors affect the lifespan of queens (*Oettler and Schrempf, 2016*). Queens that lay more eggs (total output and weekly rate) live longer than less fecund queens, irrespective of body size (*Kramer et al., 2015*), and thus seem to evade the common trade-off between reproduction and maintenance.

### Reproductive strategy
We set up 138 freshly eclosed queens from stock colonies of a Japanese population (OypB, from the Oonoyama Park in Naha, Okinawa) established in the laboratory since 2011. The experiment took place between January 2019 and January 2020. Queens were allowed to mate with a single wingless male and were placed in nest boxes with either 10, 20, or 30 workers from the maternal colony to establish monogynous colonies, for lifetime production and RNAseq (n = 46 each). These numbers of workers represent the naturally occurring number in the field and correspond to the first, median,

and third quantile of number of workers in this population (n = 62, median = 28.5, *Figure 1—figure supplement 2*). The colony was set up with half of the workers selected from inside of the nest near the brood (younger nurses) of the stock colonies, and the other half from outside the nest (older foragers) in order to minimize a putative effect of worker age on the queen (*Giehr et al., 2017*). Colonies were kept under a 12 hr dark 22°C/12 hr light 26°C cycle and fed ad libitum three times per week with diluted honey (0.6:1 honey: distilled water), cockroaches, and flies. Once per week workers, eggs, and all pupae (worker, queen, winged, and wingless male) were counted and queen survival was monitored. Additionally, the number of workers was standardized to the assigned treatment, and newly produced sexual pupae produced were removed. *C. obscurior* workers are sterile, and all produced offspring originated from the focal queen. The number of counted eggs correlates with the production of workers, queens, and the workers and queens together (*Figure 2—figure supplement 2A–C*, Kendall's rank correlation test, p<0.001: eggs-worker pupae, $\tau = 0.70$; eggs-queen pupae, $\tau = 0.59$; eggs-worker and queen pupae, $\tau = 0.73$). Pupae might have been counted more precisely than eggs, especially when larger numbers of eggs were produced. Pupae are hardly missed compared to eggs that tend to cluster together. Eggs and worker pupae might have been counted more than once as development lasts a median of 8 and 18 days for eggs and worker pupae, respectively. Colonies were counted ca. 4 weeks after the queen's death, until the last eggs had developed into pupae. Finally, three colonies (10 worker treatment) were not considered in the analysis as they were accidentally killed, leaving a total of 99 colonies for lifelong tracking and 36 colonies for RNAseq analysis.

## Worker aging

To examine worker aging, 40 focal unmarked worker pupae were set up in individual colonies with 10 or 20 marked workers (n = 20 each). These two treatments were selected because colonies with 20 and 30 workers did not differ in queen productivity. Marking of nonfocal workers was done by clipping the tarsus of the middle right leg. Colonies were set up with brood (5 larvae in the 10 workers colonies, and 10 in the 20 workers colonies), and two wingless adult queens to avoid a queenless period in case one died. The survival of the focal worker was monitored, and the number of marked workers, queens, and larvae was standardized weekly to the assigned treatment. Newly produced pupae were removed. Dead marked workers were replaced with fresh worker pupae, which were marked 1 or 2 days after eclosion to avoid confusion with the non-clipped focal worker. Dead queens were replaced with adult ones.

## Offspring investment

360 freshly eclosed adult workers were sampled monthly for head width measurements (from the third to sixth month of the queen's life, and up to five workers depending on availability). Workers were dried, pinned, and blindly measured using a Keyence Microscope at 200×. A single worker was chosen randomly and measured 10 times to obtain a proxy for measurement error (mean = 383.61 μm, standard deviation = 5.05 μm).

## Statistical tests

To test for significant differences between treatments, we used generalized linear mixed effects models within the R package glmmTMB (R version 3.5.2, *Pinheiro et al., 2011*) and a negative binomial distribution for count data. If the count data and caste investment ratios were log transformed, a Gaussian family distribution was used. The dependent variable was analyzed as a function of the fixed effects: treatment (number of workers as a factor); and random effects: stock nest and box of origin, box of set up, setup date. All models were also graphically checked for consistency and model diagnostics were performed using the DHARMA package (R version 0.3.3.0, *Hartig, 2020*). Caste ratio was calculated as queen over the total caste investment (as Queen*c / [Queen*c + Worker]). The coefficient or correction factor *c* is used as the dry average weight measurements of queen over workers to the power conversion factor of 0.7 assuming differences in metabolic rates between queens and workers adopting the logic for sex ratio investment (*Boomsma, 1989*). As this is an assumption, we used different values of *c*. The results are robust to power conversion values of 0.6–1 (*Supplementary file 1A*). To test for differences in head width, we used the average of the head width measurements of the workers per time point (each month). Predictions of the data were visualized using the loess method with the geom_smooth function and default span (ggplot2 v.3.3.2). Relative mortality and

fecundity as a function of age were mean-standardized by dividing age-specific mortality and fecundity by their means after maturation, following *Jones et al., 2014*. In contrast to *Jones et al., 2014*, the whole life range was considered until death, since removing the last 5% of survivorship showed similar results. Age-specific mortality without the mean standardization was also estimated for the 135 queens (99 and 36 ant queens for RNAseq) using a survival Bayesian trajectory analysis (*Figure 4— figure supplement 1*). Data is available as *Source data 1–Source data 7*, and the R-script used as *Source code 1*.

### *Prope mortem* queens selection

To obtain samples of low, medium, and highly productive queens, 18 queens at age 19–21 weeks were sacrificed for RNAseq based on egg productivity until week 18. Values of weekly egg productivity below the first quantile for the treatment group (colony size) were considered as low, values between the first and the third quantile as medium, and values greater than the third quantile as high. An additional 18 queens were monitored until they showed decreasing fertility (*Figure 1—figure supplement 3*) and one or more of the following signs of senescence: lethargy, loss of mobility, presence outside the nest, and/or harassment by workers. These senescent queens were also selected based on low, medium, and high fertility, and then sacrificed (28–49 weeks old). Queens were snap-frozen in liquid nitrogen after the head and thorax was separated from the gaster with a blade between the petiolus and post-petiolus in a drop of PBT 0.3% (phosphate buffered saline and Tween 20). During this procedure, queens were manipulated for less than 1 min.

### Terminology

What we refer to as 'thorax' actually refers to the thorax plus the fused first abdominal segment, together making up the 'mesosoma' in the Hymenoptera. The 'metasoma' in Hymenoptera comprises the segments making up the constriction plus the hind end. In the ant subfamily Myrmicinae, this constriction is made of two segments: the petiole corresponds to the second, constricted, abdominal segment, while the post-petiole refers to the third, constricted, abdominal segment. The 'gaster' refers to the bulbous posterior part (*Figure 1—figure supplement 5*).

### RNAseq

Total RNA was extracted using the ReliaPrep kit (Promega) from the 72 samples (36 queens, two samples per queen: head-thorax and gaster). Spike-In RNA Variant Controls (SIRV-Set 3 Lexogen #05101, Lot 5746/001492) were spiked to a 2% fraction of the total RNA (measured using Bioanalyzer – Agilent Technologies). 8 of the 72 samples showed RIN values below 7 (gaster samples from older queens that seemed more degraded). For those samples, the concentration of RNA was estimated based on the mean value of the nondegraded gaster samples. Total RNA was amplified using single primer isothermal amplification (SPIA , Ovation RNA-seq System V2, Tecan) prior to cDNA generation. The library preparation and sequencing (100 bp PE) was performed at the Cologne Center for Genomics using Nextera XT sequencing on a NovaSeq 6000 platform. Reads were trimmed with fastp v.0.20.1 to a minimum length 70 and from Nextera adapters. Then, SortMeRNA version 4.2.0 was used to discard undesired rRNA reads using the default database (smr_v4.3_default_db. fasta). Remaining reads were aligned using hisat2 (version 2.1.0) to the newest version of the genome (Cobs.2.1., *Errbii et al., 2021*). Putative splice sites were obtained using gffread (version 0.12.1), and the extensive transposable elements annotation v.2.1 (*Errbii et al., 2021*) was considered for the mapping procedure. Samtools (version 1.9) was used to sort and convert.sam into .bam files.

### Gene expression analysis

After filtering genes with 0 values, we used a gene set of 20,006 genes for the head-thorax analysis and 19,925 genes for the gaster. PCA plots were produced to visualize the samples after variance stabilizing transformation. An analysis of the homogeneity of group dispersions (variances) was performed (multivariate analog of Levene's test for homogeneity of variances), with the function permutest and 999 permutations (vegan package v. 2.5–7) to test for differences in variance among the age groups (middle-aged and *prope mortem* queens) (betadisper, vegan package).

Subsequently, a nonparametric multivariate ANOVA (PERMANOVA) test was performed (999 permutations) with the design model Head-Thorax_expression ~ AgeGroup + Fertility, with two

(middle-aged, *prope mortem*) and three levels (low, medium, and high fertility) to test for statistical differences in the transcriptomic profiles due to age group (senescent or not) and level of fertility (low, medium, and high) using the adonis function (vegan package), with the default Bray distance method. Age and fertility (average number of laid eggs per week) were scaled and centered. Then, differential expression was analyzed using the design = ~ Eggs per week + Age group, with the R package DESeq2 (v. 1.28.1). Age group was used as categorical variable with two levels (middle-aged and *prope mortem*), and log2FC were calculated as log2 [*prope mortem*/middle-aged].

The cutoff threshold of statistical significance (alpha parameter) was set as 0.001 after p-value adjustment with FDR. Functional annotation and enrichment analysis was done using topGO (version 2.46.0) and the weight01 algorithm implemented in the package.

A signed weighted co-expression network was constructed using the WGCNA package (v. 1.70–3), and the count data transformed using variance-stabilizing transformation from the DESeq2 package after excluding genes with 0 read values. For the head-thorax tissue network, the total set of 20,006 genes was used, and the WGCNA was performed with default parameters and a soft threshold power 14, based on the scale-free fit index as recommended in the manual. We compared the connectivity of the two separate networks, one for middle-aged and one for *prope mortem* queens, with the soft-Connectivity and the biweight midcorrelation functions.

## Acknowledgements

We thank Vera Ermer, Benjamin Dofka, Julia Haschlar, Lena-Marie Süß, and Judith Weber for help with the experiment, Eva Schultner, Tomer Czaczkes, Boris Kramer, Andrew Bourke, two anonymous reviewers for comments. This study was funded by the Deutsche Forschungsgemeinschaft (OE549/2-2). No funding sources were involved in study design, data collection and interpretation, or the decision to submit the work for publication.

## Additional information

### Funding

| Funder | Grant reference number | Author |
|---|---|---|
| Deutsche Forschungsgemeinschaft | OE549/2-2 | Jan Oettler<br>Jürgen Heinze |

The funders had no role in study design, data collection and interpretation, or the decision to submit the work for publication.

### Author contributions

Luisa Maria Jaimes-Nino, Data curation, Formal analysis, Investigation, Methodology, Validation, Visualization, Writing – original draft, Writing – review and editing; Jürgen Heinze, Conceptualization, Funding acquisition, Project administration, Supervision, Validation, Writing – review and editing; Jan Oettler, Conceptualization, Funding acquisition, Investigation, Methodology, Project administration, Resources, Supervision, Validation, Visualization, Writing – original draft, Writing – review and editing

### Author ORCIDs

Luisa Maria Jaimes-Nino ⓘ http://orcid.org/0000-0003-1186-1837
Jan Oettler ⓘ http://orcid.org/0000-0002-8539-6029

### Decision letter and Author response

Decision letter https://doi.org/10.7554/eLife.74695.sa1
Author response https://doi.org/10.7554/eLife.74695.sa2

## Additional files

### Supplementary files
• Supplementary file 1. Supplementary files A–I. (A) Estimates calculated for 10, 20, and 30

workers using a glmmTMB model with Gaussian distribution and 'setup date,' 'experimental box,' 'box,' and 'nest of origin' as random factors for (Queen*c)/[(Queen*c) + Worker]. The coefficient $c$ as the average dry weight of queens over workers to the power conversion factor of 0.6–1 (see text). All comparisons between 10 and 20 workers treatments are statistically significant (p<0.001), but not between 20 and 30 workers. (B) The 15 most significant enriched Gene Ontology (GO) terms per type in differentially expressed genes (DEGs) enriched in middle-aged queens in the head-thorax tissue. BP: Biological Processes; CC: cellular component; MF: molecular functions. (C) The 15 most significant enriched GO terms per type in DEGs enriched in *prope mortem* queens in the head-thorax tissue. (D) The 15 most significant enriched GO terms per type in DEGs enriched *in prope mortem* queens in the gaster tissue. (E) The 15 most significant enriched GO terms per type in DEGs enriched in middle-aged queens in the gaster tissue. (F) Shared enriched GO terms for both tissues (head-thorax and gaster) enriched in *prope mortem* queens. (G) Shared enriched GO terms for both tissues (head-thorax and gaster) enriched in middle-aged queens. (H) Deviance information criterion (DIC) of three tested models (logistic, Gompertz, and Weibull) for the age-specific mortality of ant queens. Using the function multibasta of the R package BaSTA (Survival Bayesian Trajectory Analysis, v. 1.9.5). The model with the lowest DIC value is assumed to provide the best fit. (I) Estimated coefficients of age-specific mortality of ant queens. Using a logistic model and the R package BaSTA (Survival Bayesian Trajectory Analysis, v. 1.9.5).

• Transparent reporting form

• Source data 1. Lifelong population survey of 102 queens. W indicates the treatment (number of workers) and matches the code (A and D for 30, B and E for 20, and C and F for 10 workers). Status indicates 0 = queen alive, 1 = queen dead.

• Source data 2. Weight in grams of 20 individual queens and workers.

• Source data 3. Worker head width measurements (f1) and the respective code of the maternal line (f0).

• Source data 4. Size of field colonies from Brazil collected in 2013 (BR13) and 2018 (BR18) and Japan in 2011 (JP11) from *Schrader et al., 2014*.

• Source data 5. Samples for RNA sequencing and experimental design information.

• Source data 6. Population survey of 36 queens. W indicates the treatment (number of workers) and matches the code (A and D for 30, B and E for 20, and C and F for 10 workers). Status indicates 0 = queen alive, 1 = queen's sacrifice.

• Source data 7. Survival probability of queens over time.

• Source data 8. Lifelong survey of 40 workers in 40 independent colonies.

• Source data 9. Differentially expressed genes by age in gaster. 4832 of 19,925 expressed genes were differentially expressed between age groups (after false discovery rate [FDR] adjustment p<0.001, DESeq2).

• Source data 10. Differentially expressed genes by age in head-thorax. 3565 genes were differentially expressed between middle-aged and *prope mortem* queens (after false discovery rate [FDR] adjustment p<0.001, DESeq2) out of 20,006 expressed genes.

• Source code 1. R-script used to analyze the population survey data.

## Data availability

Data generated or analysed during this study are included in the manuscript and supporting file; A source Data file has been provided for Figures 1–5 and accompanying figure supplements. Raw sequencing data have been deposited in SRA under the BioProject accession number PRJNA819887. The Bioproject is part of the NCBI So-Long Umbrella Bioproject: "The So-Long project: Sociality and the Reversal of the Fecundity/Longevity Trade-off.

The following dataset was generated:

| Author(s) | Year | Dataset title | Dataset URL | Database and Identifier |
|---|---|---|---|---|
| Jaimes-Nino L | 2022 | Cardiocondyla obscurior Raw sequence reads (TaxID: 286306) | https://www.ncbi.nlm. nih.gov/bioproject/? term=PRJNA819887 | NCBI BioProject, PRJNA819887 |

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
