## [Editor Report]

The article will contribute significantly to our understanding of superorganism development and longevity in ants and will provide testable hypotheses in other ant species and other organisms.

---

## [Decision Letter]

**Decision letter after peer review:**

Thank you for submitting your article "Late fitness gains explain the delay of the selection shadow in ants" for consideration by *eLife*. Your article has been reviewed by 2 peer reviewers, and the evaluation has been overseen by a Reviewing Editor and Claude Desplan as the Senior Editor. The reviewers have opted to remain anonymous.

Essential revisions:

The editors and both reviewers agree that this dataset has potentially important implications for understanding superorganism development and the long lifespans of social insects. Therefore, it will be of interest to a broad audience of biologists. Both reviewers have substantive comments on data interpretation and writing, as well as requests for some additional information and statistical analyses. Specifically:

1) Data interpretation:

a) A particular concern regarding the interpretation of the current data set is that it does not necessarily provide an ultimate explanation for the long life of the queens: an alternative, more conservative, interpretation of the current data set is that it only provides documentation of negligible reproductive senescence. According to Reviewer 2, the long lifespans of queens is strongly correlated too, but is not necessarily caused by, the capacity to reproduce late in life. For example, the long lifespans of queens could be ultimately caused by a shifting of reproductive costs to workers, or by living in protected environments. Therefore, in Reviewer 2's view, the current data set can only address the question of why social insect queens lack reproductive senescence. In other words, why do social insect queens retain the capacity to reproduce late in life? In my view, you (the Authors) should be very careful to distinguish ultimate from proximate causes in this study and be very clear about the precisely question this current data set can or cannot address.

b) Following on point (a) above, Reviewer 3 raises another particular point of concern, which is the lack of consideration for the alternative hypothesis that the drop in fertility later in life is caused by sperm limitation and not reproductive senescence. As Reviewer 3 points out, unless you can rule out this alternative hypothesis, it must be explicitly considered in the main text and would change the interpretation of the data, but not necessarily the relevance of the study, especially if you are willing shift the focus instead on understanding intrinsic properties of superorganism development and less on ageing per se.

c) Another issue of particular concern is the unnecessary focus on queen control. The inflection point, where colonies produce more queen offspring late in life, appears to reflect an intrinsic property of colony (superorganism) development that is independent of the colony-size, lifespan or worker investment. How this inflection is mechanistically controlled is a question you should address in the Discussion and should offer several hypotheses. For example, three possible hypotheses are: (1) An intrinsic colony-level threshold that is be controlled by queens and workers, such that workers change their feeding/foraging behavior in response to a queen pheromone once the queen reaches a certain age; (2) Queen embryos are produced all the time, like in the genus Monomorium, but are killed by workers until the queen reaches a certain age and changes its pheromone status; (3) as you have suggested, the threshold could be genetically determined in the queen regardless of worker input. Therefore, you should remove the focus on the assumption of queen control and provide a more systems-level (superorganismal) explanation for the inflection for producing queen offspring independent of colony size in the discussion.

2) RNAseq data: If you feel that RNAseq data can strengthen your arguments in light of the comments / critiques on data interpretation raised above, then you should add these data to the manuscript. If you decide to do so, please make sure to explain why this is important both in the cover letter and in the response to reviewers.

3) Reviewers 2 and 3 require some missing statistics as well as analyses, which should be straight forward to perform.

4) The manuscript, as presently written, leaves many open questions that need clarification, therefore there is a need for major revision to respond to the queries of Reviewer 2 and 3 (see Reviewer comments below).

5) You should provide additional information about this species' life history (see Reviewer comments below).

*Reviewer #1 (Recommendations for the authors):*

This is an interesting study looking at the evolution of ageing in social insects using ants as a model. As I haven't seen the initial submission, I have looked at the manuscript and the response to reviewers and I base my suggestions on both documents.

Evolution of ageing remains only partially understood and this field seems to be experiencing a sort of renaissance in recent years with a surge of theoretical advances and new empirical findings. Queens of social insects, and ant queens in particular, have remarkable lifespans and understanding the biology of their long life can help in understanding the biology of ageing in a more general sense.

In this study, the authors focus on following quite a large number of ant (C. obscurior) colonies and provide intriguing data in relation to age-specific mortality and reproduction. The gist of their argument is that the mortality is decreasing with age while reproduction (production of sexuals) is increasing with age, such that there is little evidence of ageing in this species.

Overall, I think this is an interesting dataset that provides important information that will advance the field. However, I think the manuscript currently lacks clarity, structure and suffers from poor formulation of ideas in places, and is rather difficult to follow even for an expert in the field. I think that it requires quite a bit of work to sort this out; nevertheless, I think the dataset is, potentially, sufficiently strong to justify publication in a high-impact journal such as *eLife* following major revision. However, I also have a methodological question (#15) which could be key for the interpretation of the results

I will provide comments in a sequential order, as they appeared while reading the manuscript.

1. You must include the species name either in the title or in the abstract. Currently it reads as if you followed 102 colonies of random species of ants.

2. L 21 advanced not advance.

3. "Furthermore, mortality decreased with age in queens and workers, supporting the hypothesis that aging trajectories are adaptive." I didn't understand this sentence while reading the abstract and reading the reminder of the paper didn't make clearer. What is meant here? Why would decrease in mortality suggest that "aging trajectories" are adaptive? What are "aging trajectories"?

4. "We argue that selection for late life reproduction delays the occurrence of the selection shadow, leads to the apparent absence of senescence in ants and underlies the evolution of long lifespans." This is very confusing – if "selection shadow" is delayed, how come there is an apparent absence of senescence? "Delayed" suggests that senescence will occur but at a later point. Also, this sentence is a bit clumsy and contains grammatical errors.

5. "Together, this highlights the unique aging patterns of social insects in comparison to other iteroparous species across the tree of life." Why? Is there evidence that negligible senescence is unique characteristic of social insects? Also, "aging patterns" is unclear.

6. Introduction: I found it very confusing that the paper on ageing starts with the discussion of reproductive queens and workers of social insects. You need a general introduction paragraph. The most straightforward solution is to take the first paragraph from Conclusions (LL 138-151) and start your main text with this.

7. Introduction: "It is argued that the costs of reproduction (energy intake, brood care) are outsourced to the workers, but this is a necessary and not sufficient condition for the evolution of long lifespans in social insects and does not explain these aging patterns." Why? There is no explanation as to why this is not sufficient, and it is absolutely not clear to me – please explain.

8. LL 45-58 – I like this paragraph and I think it presents good arguments in favour of the novelty provided by this study

9. L 58 – "aging, senescence" – if you treat ageing and senescence as different terms, you have to explain why and provide definitions for each term.

10. L 58 – "investment" – what investment? Reproduction? Somatic maintenance? This is sentence is remarkably unclear.

11. I think that "superorganism" approach, while still debated in the literature is entirely justified in this system.

12. L81 – While not directly relevant as you did not find a correlation, how did you test for causality of correlation? Your experimental treatment arguably had no effect neither on egg production nor on queen lifespan?

13. "Generalizing this finding, we suggest that lat*eLife* investment and its effect on queen aging is an intrinsic property of superorganismality irrespective of the mode of caste determination and social organization." – this is unclear, I thought there is no effect on queen's ageing?

14. I think you should analyse age-specific mortality by selecting the best-fit model in a specialized demography analyses package, such as, for example, BASTA.

15. My understanding is that queens live for 40-50 weeks max (Figure S3). Figure 4 suggests that from week 30 onwards the production of eggs, worker pupae and queen pupae declines. This suggests that while queen mortality declines in late life, so does queen reproduction. So, queens of this species do show reproductive senescence?

Yes, your data suggests that relative investment into reproduction (queen worker ratio) increases with age, but the absolute number of queens declines with age? To me, this suggests an interesting result from the life-history theory perspective – increased investment in reproduction with reduced residual reproductive value, but not necessarily the absence of reproductive senescence. Perhaps I didn't understand the results, please clarify.

16. L155 – this relative mortality is very unclear to me. I think you need proper mortality rate analyses in BASTA and estimate different age-specific mortality parameters.

17. L 156 'random' offspring?

18. "In turn, the short selection shadow might have led to the delay of a distinguishable senescence phase." I think I understand what you mean but this is very unclear, please re-write.

*Reviewer #2 (Recommendations for the authors):*

Line 42: I'd like clarification on your argument here. Why is this not sufficient in your opinion? I don't see any alternative explanation, and workforce dynamics is central to your experimental design. After all, you are manipulating workforce size to expose trade-offs in queen decisions, so how does it not ultimately come down to the workforce?

Line 72: Why not just say something simple like "compare queen and worker mortality". Throwing in a new term like aging trajectories makes it seem like you are doing something different.

Line 92: I would have liked to see more of your data on the male production, even if they were less common than queens, 15% of your reproductive output is not negligible. Does male production fit the same lat*eLife* pattern as queen production? Do you see the same differences between treatments when you analyze combined alate production? I expect the 10-worker treatment may invest more in males, since they are usually cheaper to produce. I know this isn't the question you are asking but it may explain some of the drop in queen production in that treatment.

Line 154: I don't understand why you seem to downplay the clear influence that the workforce has on this dynamic. From my perspective, your data beautifully shows how these queens transition from worker production to alate production as they age, and that colonies with low worker numbers cannot invest as much into reproduction, which is expected in a species that must establish a robust colony before devoting resources to reproduction. If a queen survives to a relative old age, she will also have a large, established workforce that can raise a lot of alates, so of course we will see longevity selected for at that point (as indicated by your own data, if she reaches that age without a sufficient workforce, her alate production will suffer).

Another important omission in the conclusion is the inevitable problem of sperm limitation. You frame the system as generating a directional push towards negligible senescence but it is more likely stabilizing selection that produces a sufficient workforce and then devotes as much remaining sperm as possible towards queen production. Selection will ensure that an established queen lives long enough to exhausts her sperm storage, which would contribute to the short selection shadow you discuss. The increase in relative mortality at 30 weeks may just represent the optimal lifespan of the colony. In figure 2A we see the peak in queen production at that time point, with both worker and queen production falling dramatically by 40 weeks, potentially signaling sperm depletion. Upon closer inspection of S3, it looks like there were only a few colonies that survived past week 38, but then persisted for another couple months. I have to wonder how reliable the drop in relative mortality is after 40 weeks when it represents only a few colonies. Of course, this is my interpretation based off of the figures, so please include the raw numbers if I am mistaken.

Line 212: Is this species monogamous in the wild? If so, this should be stated because if they are normally polyandrous, having only one mate may introduce sperm limitation.

[Editors' note: further revisions were suggested prior to acceptance, as described below.]

Thank you for resubmitting your work entitled "Lat*eLife* fitness gains and reproductive death in *Cardiocondyla obscurior* ants" for further consideration by *eLife*. Your revised article has been evaluated by Claude Desplan (Senior Editor) and a Reviewing Editor. The paper is significantly improved but we believe that additional editorial changes must be made to increase the impact of the paper, as described below, before the paper can be accepted for publication.

Although the transcriptomic data were analyzed properly, the conclusions that can be drawn from these data are limited, primarily because the difference between gaster and head/thorax is not easy to interpret. However, we think that these data do add something to the paper by validating some of the conclusions drawn from indirect observations. Therefore, the RNAseq do not necessarily serve the purpose of identifying which mechanisms are failing, but instead provide evidence that the queens are experiencing delayed senescence, something that was not obvious in the previous version. Please keep the interpretation of RNAseq simple and revise the manuscript appropriately to make its purpose clearer (see comment below on Line 82-83 and 178 and 179 below).

Furthermore, the singular focus on the queen is problematic, especially if you are attempting to adopt a superorganism perspective. First, the evidence presented that only the queen is in control is indirect and weak. Therefore, we ask you to maintain a neutral stance on this issue and acknowledge that the workers may play an important role in this process. Second, your arguments do not depend on whether determination is in the maternal, embryonic, or larval stage and we ask you to remove this mention as it is a non-central point in your argument (lines 127-130). Given these two points, we ask you to put back the data on worker longevity (although it can be presented in the supplemental data), because in our view, without direct supporting evidence that it is all about the queen, workers remain an important part of the equation. We therefore ask you to revise lines 127-130 in the Intro and 287-297 in the Conclusions section.

Essential revisions:

Abstract:

Line 23-25: We ask you to add something to describe the inflection point as a threshold, an inflection point that changes in an abrupt, non-linear-like way which distinguishes it from a gradual linear-like increase.

What does "reproductive death" mean?

Introduction:

Line 46: What does "loss of function" refer to specifically?

Line 49: Clarify what "Among others" is referring to here. We assume it means explanations / theories

Lines 55-73: Should also cite: Tschinkel, Insect. Soc. (2017) 64:285-296: DOI 10.1007/s00040-017-0544-0

Line 75: Cite original Wheeler (1911) paper (and if appropriate the Holldobler and Wilson book The Superorganism).

Line 82-83: The RNAseq do not necessarily serve the purpose of identifying which mechanisms are failing, but instead provide evidence that the queens are experiencing delayed senescence. Something that was not obvious in the previous version. Therefore, this should be mentioned in the manuscript.

Short methods:

Integrate into Intro, it's not part of the *eLife* format to have a "Short Methods" section and shorten it.

Results:

Line 125-126: Please clarify the meaning of "This confirms a previous study, which did not find a causal correlation between queen's lifespan and fecundity (Schrempf et al. 2017)." We are not clear how manipulation of # of workers in the colony leads to this specific conclusion. The result can be interpreted more conservatively as "no relation between queen's lifespan and colony size."

Lines 127-130: As in the previous round of revision, this is not necessary for the impact of the paper and we ask you to remove this, especially in light in the Conclusion section on Lines 281-286, stating it is not important whether it occurs maternally, during embryogenesis, larval development, etc …

Lines 140-141: Here is an opportunity to rule out the sperm depletion hypothesis. It is also redundant with Lines 172-174

Line 82and83 and 177and178: please make the question RNAseq analysis explicit. The question of "how" the queens are dying cannot be answered solely by RNaseq. Instead, focus the question on responding to Reviewer 2/3's critique, which is to distinguish between sperm limitation and senescence and to confirm that there is both reproductive and body senescence. Would we have expected that both head-thorax and gaster to both senesce? We would have expected that head-thorax but not reproductive tissue to senesce. Therefore making the question explicit and some discussion of the predictions and findings would improve this section considerably.

Finally, how confident are we that the expression changes presented really indicate ageing. I would have appreciated more detailed justification. For example, you could include a table with the functions of key genes that are differentially expressed, and then we could see what down or up regulation in old queens would mean. Alternatively, you could choose a couple of examples in the text and expand upon them. For example, we understand that gene co-expression networks are associated with aging, but why? This is a key result that distinguishes thorax-head and gaster transcriptome and is central to your thesis later that thorax-head deteriorates physiologically while gaster maintains some functionality. Therefore, some more justification would allow the reader to better appreciate these changes. See also comment on Lines 247-249.

Discussion:

Abstract and Line 246: Please remove the word "dramatic" before physiological changes, since you only have a 2-point and not 3-point comparison.

Lines 247-249: This statement is not well supported since they were not statistically different: "The changes are stronger in the gaster, which contains the reproductive organs and most of the digestive system, but to a similar extent occur in head and thorax, containing most neuronal and muscle tissue." Please refer explicitly to the evidence supporting this statement.

Line 251-252: Please provide some examples of these genes.

Conclusion:

Lines 287-297: Once again, these predictions are placing singular focus on the queen, without taking workers into account, which in our view, is antithetical to the superorganism perspective. Without direct supporting evidence, you cannot rule out that the interaction between queens and workers is just as important to the superorganism perspective as is the queen herself. In our view, we ask you to maintain a neutral stance on this point and ask that you maintain the data on worker longevity that you removed in the previous version and revise the text.

---

## [Author Response]

Essential revisions:The editors and both reviewers agree that this dataset has potentially important implications for understanding superorganism development and the long lifespans of social insects. Therefore, it will be of interest to a broad audience of biologists. Both reviewers have substantive comments on data interpretation and writing, as well as requests for some additional information and statistical analyses. Specifically:1) Data interpretation:a) A particular concern regarding the interpretation of the current data set is that it does not necessarily provide an ultimate explanation for the long life of the queens: an alternative, more conservative, interpretation of the current data set is that it only provides documentation of negligible reproductive senescence. According to Reviewer 2, the long lifespans of queens is strongly correlated too, but is not necessarily caused by, the capacity to reproduce late in life. For example, the long lifespans of queens could be ultimately caused by a shifting of reproductive costs to workers, or by living in protected environments. Therefore, in Reviewer 2's view, the current data set can only address the question of why social insect queens lack reproductive senescence. In other words, why do social insect queens retain the capacity to reproduce late in life? In my view, you (the Authors) should be very careful to distinguish ultimate from proximate causes in this study and be very clear about the precisely question this current data set can or cannot address.

The long lifespans of queens can be explained ultimately as a consequence of higher fitness returns late in life, and could be caused proximately by shifting costs to the workers. This is shown by our study, as queens with 30 workers are capable of rearing more sexuals than queens with 10 workers. Additionally, it is now known that living in a more protected environment does not mean a decrease in mortality (see Moorad Trends Ecol Evol 2020).

We now make clear that aging has two dimensions following Baudisch (2011) and Baudisch and Scott (2019) (now in Line 60). The first dimension “pace” refers to factors describing the time-scale (i.e. the long lifespans of reproductive of social insects). By doing so, we explain that this dataset was not obtained to answer directly questions related to the pace of aging. Instead, this dataset was obtained to answer questions related to the “shape” of aging. So, how is relative mortality and fertility distributed along the ant queens’ life, and how this directly affects the age-specific selection strength. Nevertheless, this is a fundamental start to understand the “pace” of aging. Hopefully with this we now make clear to the readers about the reach and limitations of this study.

b) Following on point (a) above, Reviewer 3 raises another particular point of concern, which is the lack of consideration for the alternative hypothesis that the drop in fertility later in life is caused by sperm limitation and not reproductive senescence. As Reviewer 3 points out, unless you can rule out this alternative hypothesis, it must be explicitly considered in the main text and would change the interpretation of the data, but not necessarily the relevance of the study, especially if you are willing shift the focus instead on understanding intrinsic properties of superorganism development and less on ageing per se.

In line 188 we explain that sperm limitation is not a plausible explanation for the drop of egg production late in life, as we only notice a final increase of solely male pupae production at the end of life in one colony. Queens were mated with wingless males which should transfer an excess amount of sperm (Schrempf and Heinze, 2008), and we show a high production of diploid queen pupae at the end of life.

c) Another issue of particular concern is the unnecessary focus on queen control. The inflection point, where colonies produce more queen offspring late in life, appears to reflect an intrinsic property of colony (superorganism) development that is independent of the colony-size, lifespan or worker investment. How this inflection is mechanistically controlled is a question you should address in the Discussion and should offer several hypotheses. For example, three possible hypotheses are: (1) An intrinsic colony-level threshold that is be controlled by queens and workers, such that workers change their feeding/foraging behavior in response to a queen pheromone once the queen reaches a certain age; (2) Queen embryos are produced all the time, like in the genus Monomorium, but are killed by workers until the queen reaches a certain age and changes its pheromone status; (3) as you have suggested, the threshold could be genetically determined in the queen regardless of worker input. Therefore, you should remove the focus on the assumption of queen control and provide a more systems-level (superorganismal) explanation for the inflection for producing queen offspring independent of colony size in the discussion.

As here suggested, the exact mechanism of determination is not necessary for the discussion of the results. We refrain from discussing the three hypotheses, as we are sure that caste is determined in the embryo. In addition, an older study (Suefuji et al., 2008 Biol Lett.) showed that queens not only produce males faster in multiple queen colonies, but also female sexuals. This was not discussed in that paper, but also points to queen control. Nevertheless, we state clearly that this is not the interesting point, but whether queen and worker interests are aligned (Line 310).

We have a manuscript in preparation that suggests that workers do not discriminate between queen and worker larvae, and that these are also chemically not distinguishable. This supports that interests are aligned.

On a sidenote: Queen larvae in *Monomorium* are usually culled in the presence of a queen. In our species, and in this experiment, queens are produced all the time.

2) RNAseq data: If you feel that RNAseq data can strengthen your arguments in light of the comments / critiques on data interpretation raised above, then you should add these data to the manuscript. If you decide to do so, please make sure to explain why this is important both in the cover letter and in the response to reviewers.

We have included RNASeq data of 36 queens. These ant queens belonged to the same population survey. The data compare fit queens with queens close to death, and is a substantial extension to the study. Specifically, this is a fundamental point in the discussion of how selection strength is maintained until the sexual investment peak is reached in ant queens.

3) Reviewers 2 and 3 require some missing statistics as well as analyses, which should be straight forward to perform.

We have addressed this point in the revised version.

4) The manuscript, as presently written, leaves many open questions that need clarification, therefore there is a need for major revision to respond to the queries of Reviewer 2 and 3 (see Reviewer comments below).

Responded below.

5) You should provide additional information about this species' life history (see Reviewer comments below).

We have included a “Short methods” section (Line 94-127) which now includes a small section about the species’ life history.

Reviewer #1 (Recommendations for the authors):[…]I will provide comments in a sequential order, as they appeared while reading the manuscript.1. You must include the species name either in the title or in the abstract. Currently it reads as if you followed 102 colonies of random species of ants.

Now the species name is included in the title and abstract.

2. L 21 advanced not advance.

Done.

3. "Furthermore, mortality decreased with age in queens and workers, supporting the hypothesis that aging trajectories are adaptive." I didn't understand this sentence while reading the abstract and reading the reminder of the paper didn't make clearer. What is meant here? Why would decrease in mortality suggest that "aging trajectories" are adaptive? What are "aging trajectories"?

To avoid confusion with the terminology we have decided to use the terminology in Baudisch (2001) and Baudosch and Scott (2019) about aging as the consequence of “pace” and “shape” of demographic trajectories (mortality and fertility), now clarified in Line 60. This part has also changed, as we now calculated relative mortality as a function of age.

4. "We argue that selection for late life reproduction delays the occurrence of the selection shadow, leads to the apparent absence of senescence in ants and underlies the evolution of long lifespans." This is very confusing – if "selection shadow" is delayed, how come there is an apparent absence of senescence? "Delayed" suggests that senescence will occur but at a later point. Also, this sentence is a bit clumsy and contains grammatical errors.

We agree and we have reformulated the abstract. We meant that senescence is apparently absent, as it can only be perceived in a short window of time shortly before death. We have changed this throughout the manuscript.

5. "Together, this highlights the unique aging patterns of social insects in comparison to other iteroparous species across the tree of life." Why? Is there evidence that negligible senescence is unique characteristic of social insects? Also, "aging patterns" is unclear.

We have addressed this now in the introduction, in which we explain that aging theory predicts that senescence has its onset after maturity, and this is seen in iteroparous species. Here we show that the onset of senescence in ant queens is past that point and is delayed.

6. Introduction: I found it very confusing that the paper on ageing starts with the discussion of reproductive queens and workers of social insects. You need a general introduction paragraph. The most straightforward solution is to take the first paragraph from Conclusions (LL 138-151) and start your main text with this.

Aging and reproduction are life-history traits that are interconnected. Now we start with two general paragraphs that bring up both terms.

7. Introduction: "It is argued that the costs of reproduction (energy intake, brood care) are outsourced to the workers, but this is a necessary and not sufficient condition for the evolution of long lifespans in social insects and does not explain these aging patterns." Why? There is no explanation as to why this is not sufficient, and it is absolutely not clear to me – please explain.

It has been argued that, if costs are not overtaken by workers (in terms of protection, energy intake, etc), this will result in increasing levels of extrinsic mortality experienced by ant queens. And this has long been used as an explanation for the long lifespans of social insects. But as already explained by Hamilton and later recapitulated by Moorad and colleagues (now in line 251), age-independent extrinsic mortality cannot explain the pace of aging. Therefore, this cannot be a sufficient explain on why social insects live so long. Additionally, our study shows that an increase in worker force does not affect queens’ lifespan, confirming Moorad et al.

8. LL 45-58 – I like this paragraph and I think it presents good arguments in favour of the novelty provided by this study

Thanks.

9. L 58 – "aging, senescence" – if you treat ageing and senescence as different terms, you have to explain why and provide definitions for each term.

Now we define aging in line 44, and senescence in line 37.

10. L 58 – "investment" – what investment? Reproduction? Somatic maintenance? This is sentence is remarkably unclear.

Now we specify reproductive investment (Line 78).

11. I think that "superorganism" approach, while still debated in the literature is entirely justified in this system.12. L81 – While not directly relevant as you did not find a correlation, how did you test for causality of correlation? Your experimental treatment arguably had no effect neither on egg production nor on queen lifespan?

We did not test for causality of correlation, we did not find an effect of our treatment in the egg productivity nor queen lifespan.

13. "Generalizing this finding, we suggest that lateLife investment and its effect on queen aging is an intrinsic property of superorganismality irrespective of the mode of caste determination and social organization." – this is unclear, I thought there is no effect on queen's ageing?

We explain in Line 310 that the mode of determination is not relevant. What is important is whether queen and worker interests are aligned, so whether there is conflict over sex and caste of the offspring. This can of course shape the pace of aging.

14. I think you should analyse age-specific mortality by selecting the best-fit model in a specialized demography analyses package, such as, for example, BASTA.

We prefer to show the relative mortality as a function of age as in Jones et al. (2014), as this gives valuable information about the specific age at which mortality increases over average life-time mortality (when mortality passes the threshold of 1 in Figure 4.) As suggested by the reviewer we also now include age-specific mortality of the best-model fitted using BaSTA and the estimated parameters in the supplement (Figure 4 —figure supplement 1, Supplementary File 1H and 1I), which does not add additional insights.

15. My understanding is that queens live for 40-50 weeks max (Figure S3). Figure 4 suggests that from week 30 onwards the production of eggs, worker pupae and queen pupae declines. This suggests that while queen mortality declines in late life, so does queen reproduction. So, queens of this species do show reproductive senescence?

Yes, now better shown in Figure 2 —figure supplement 1.

Yes, your data suggests that relative investment into reproduction (queen worker ratio) increases with age, but the absolute number of queens declines with age? To me, this suggests an interesting result from the life-history theory perspective – increased investment in reproduction with reduced residual reproductive value, but not necessarily the absence of reproductive senescence. Perhaps I didn't understand the results, please clarify.

Yes, we agree with you. We hope now it becomes clear from the text that queens do experience reproductive senescence.

16. L155 – this relative mortality is very unclear to me. I think you need proper mortality rate analyses in BASTA and estimate different age-specific mortality parameters.

See comment 14.

17. L 156 'random' offspring?

The word ‘random’ has been removed as it did not add new information.

18. "In turn, the short selection shadow might have led to the delay of a distinguishable senescence phase." I think I understand what you mean but this is very unclear, please re-write.

We have rewritten the respective parts. The RNAseq data reveal a clear senescence phase, evident at the end of life and not in middle aged queens. Clearly, only a time course analysis can reveal whether senescence occurs earlier, the fecundity data however speak against a prolonged phase of senescence.

Reviewer #2 (Recommendations for the authors):Line 42: I'd like clarification on your argument here. Why is this not sufficient in your opinion? I don't see any alternative explanation, and workforce dynamics is central to your experimental design. After all, you are manipulating workforce size to expose trade-offs in queen decisions, so how does it not ultimately come down to the workforce?

As answered in the point 7 for reviewer 1: It has been argued that, if costs are not overtaken by workers (in terms of protection, energy intake, etc), this will result in increasing levels of extrinsic mortality experienced by ant queens. And this has been long used as an explanation for the long lifespans of social insects, but as explained by Hamilton and later recapitulated by Moorad and colleagues (now in line 251), age-independent extrinsic mortality cannot explain the pace of aging. Therefore, this cannot be a sufficient explanation why social insects live so long. Additionally, our study shows that an increase in worker force does not affect queens’ lifespan.

Line 72: Why not just say something simple like "compare queen and worker mortality". Throwing in a new term like aging trajectories makes it seem like you are doing something different.

We have removed the worker data in order to focus on the queen’s data.

Line 92: I would have liked to see more of your data on the male production, even if they were less common than queens, 15% of your reproductive output is not negligible. Does male production fit the same lateLife pattern as queen production? Do you see the same differences between treatments when you analyze combined alate production? I expect the 10-worker treatment may invest more in males, since they are usually cheaper to produce. I know this isn't the question you are asking but it may explain some of the drop in queen production in that treatment.

We have now included relative male production in Figure 4, and it follow the same late pattern as queen production. The lifetime production of wingless males is very low (mean: 0.36, median: 0, N = 99 queens). We did not test for winged males (there is a male polyphenism in *Cardiocondyla*) specifically as there were almost no produced. Further, it is not clear whether wingless males are low-cost investments, as males are under strong sexual selection, adapted for fighting with rivals, have longish lifespans and lifelong spermatogenesis.

Line 154: I don't understand why you seem to downplay the clear influence that the workforce has on this dynamic. From my perspective, your data beautifully shows how these queens transition from worker production to alate production as they age, and that colonies with low worker numbers cannot invest as much into reproduction, which is expected in a species that must establish a robust colony before devoting resources to reproduction. If a queen survives to a relative old age, she will also have a large, established workforce that can raise a lot of alates, so of course we will see longevity selected for at that point (as indicated by your own data, if she reaches that age without a sufficient workforce, her alate production will suffer).

We have now included in Line 262 “The magnitude of such investment (i.e. number of queen pupae produced) is affected by the number of workers available”, to acknowledge the role of worker size, so this message is not lost from the discussion.

Another important omission in the conclusion is the inevitable problem of sperm limitation. You frame the system as generating a directional push towards negligible senescence but it is more likely stabilizing selection that produces a sufficient workforce and then devotes as much remaining sperm as possible towards queen production. Selection will ensure that an established queen lives long enough to exhausts her sperm storage, which would contribute to the short selection shadow you discuss. The increase in relative mortality at 30 weeks may just represent the optimal lifespan of the colony. In figure 2A we see the peak in queen production at that time point, with both worker and queen production falling dramatically by 40 weeks, potentially signaling sperm depletion. Upon closer inspection of S3, it looks like there were only a few colonies that survived past week 38, but then persisted for another couple months. I have to wonder how reliable the drop in relative mortality is after 40 weeks when it represents only a few colonies. Of course, this is my interpretation based off of the figures, so please include the raw numbers if I am mistaken.

The decrease of queen and worker production due to sperm limitation is highly unlikely as we only found one case of sperm depletion in the whole data set (now reported in line 154). In the new version of the manuscript, we have included that in this species queens generally mate once, and that ergatoid males transfer an excess amount of sperm making very uncommon the cases of sperm depletion.

After revision of the relative mortality as a function of age according to Jones et al. 2014, we find an increase of mortality above average after 30 weeks in ant queens, corroborated by the suggested BaSTA analysis. Additionally, raw numbers of exact number of surviving queens at each age are reported in the Source Data 7.

Line 212: Is this species monogamous in the wild? If so, this should be stated because if they are normally polyandrous, having only one mate may introduce sperm limitation.

We have now included this information in Line 98 as it follows: “Virgin queens usually mate once with related wingless males inside the natal colony (Heinze and Hölldobler 1993; Schmidt et al. 2016; Heinze and Hölldobler 2019), generally stay in the nest, and new colonies are formed by budding of colony fragments.”

[Editors' note: further revisions were suggested prior to acceptance, as described below.]

Although the transcriptomic data were analyzed properly, the conclusions that can be drawn from these data are limited, primarily because the difference between gaster and head/thorax is not easy to interpret. However, we think that these data do add something to the paper by validating some of the conclusions drawn from indirect observations. Therefore, the RNAseq do not necessarily serve the purpose of identifying which mechanisms are failing, but instead provide evidence that the queens are experiencing delayed senescence, something that was not obvious in the previous version. Please keep the interpretation of RNAseq simple and revise the manuscript appropriately to make its purpose clearer (see comment below on Line 82-83 and 178 and 179 below).

We realize that there are problems with the data. For one, as you point out, it is futile to speculate which processes fail in old queens, as there are many. Second, we only scratch the surface in a non-model insect and rely on assumptions. Third, we only compare two time points, thus ‘changes’ could be transient. Lastly, we deduce a pattern from two different studies. Harrison et al. (2020) compare young with middle-aged queens just before their reproductive peak and find no signs of senescence, rather the contrary. Here, we compare middle-aged at their reproductive peak with *prope mortem* queens and show strong signs senescence. We hope to have revised the manuscript accordingly to make the purpose of the RNAseq data clear.

Furthermore, the singular focus on the queen is problematic, especially if you are attempting to adopt a superorganism perspective. First, the evidence presented that only the queen is in control is indirect and weak. Therefore, we ask you to maintain a neutral stance on this issue and acknowledge that the workers may play an important role in this process. Second, your arguments do not depend on whether determination is in the maternal, embryonic, or larval stage and we ask you to remove this mention as it is a non-central point in your argument (lines 127-130). Given these two points, we ask you to put back the data on worker longevity (although it can be presented in the supplemental data), because in our view, without direct supporting evidence that it is all about the queen, workers remain an important part of the equation. We therefore ask you to revise lines 127-130 in the Intro and 287-297 in the Conclusions section.

We have changed the respective section. Please understand, that coming from a background nested in the vast diversity of myrmecological life, the last author struggles with generalizations. Indeed, the focus is unnecessary and does not add to the argument.

Essential revisions:Abstract:Line 23-25: We ask you to add something to describe the inflection point as a threshold, an inflection point that changes in an abrupt, non-linear-like way which distinguishes it from a gradual linear-like increase.

We now describe the non-linear increase of sexual production: “By life-long tracking of 99 *Cardiocondyla obscurior* (Formicidae: Myrmicinae) ant colonies, we find that queens shift to the production of sexuals in late life regardless of their absolute lifespan or worker investment”

What does "reproductive death" mean?

We now state that it is the development of massive pathology following reproductive effort as:

“Furthermore, RNAseq analyses of old queens past that fitness peak showed the development of massive pathology while still being fertile, leading to rapid post-reproductive death.”

Introduction:Line 46: What does "loss of function" refer to specifically?

Now should be clear as “…and may lead to loss of reproductive performance and survival (i.e. senescence).”

Line 49: Clarify what "Among others" is referring to here. We assume it means explanations / theories.

Done.

Lines 55-73: Should also cite: Tschinkel, Insect. Soc. (2017) 64:285-296: DOI 10.1007/s00040-017-0544-0.

Done.

Line 75: Cite original Wheeler (1911) paper (and if appropriate the Holldobler and Wilson book The Superorganism).

Done.

Line 82-83: The RNAseq do not necessarily serve the purpose of identifying which mechanisms are failing, but instead provide evidence that the queens are experiencing delayed senescence. Something that was not obvious in the previous version. Therefore, this should be mentioned in the manuscript.

We included now the purpose of the RNAseq analysis to provide evidence of senescence in this *prope mortem* queens as:

“To assess if senescence was restricted particularly to the end of life we compared RNAseq data of 18 of such *prope mortem* (Lat. near death) queens (between 28-29 weeks old) and 18 middle-aged queens (between 19-21 weeks) which were in their peak of fertility (Figure 1 —figure supplement 3A-B.)”

Short methods:Integrate into Intro, it's not part of the eLife format to have a "Short Methods" section and shorten it.

Done.

Results:Line 125-126: Please clarify the meaning of "This confirms a previous study, which did not find a causal correlation between queen's lifespan and fecundity (Schrempf et al. 2017)." We are not clear how manipulation of # of workers in the colony leads to this specific conclusion. The result can be interpreted more conservatively as "no relation between queen's lifespan and colony size."

We decided to remove this sentence as it does not add to the discussion.

Lines 127-130: As in the previous round of revision, this is not necessary for the impact of the paper and we ask you to remove this, especially in light in the Conclusion section on Lines 281-286, stating it is not important whether it occurs maternally, during embryogenesis, larval development, etc …

We deleted those lines.

Lines 140-141: Here is an opportunity to rule out the sperm depletion hypothesis. It is also redundant with Lines 172-174

We deleted lines 172-174 and left the ones in the result section.

Line 82and83 and 177and178: please make the question RNAseq analysis explicit. The question of "how" the queens are dying cannot be answered solely by RNaseq. Instead, focus the question on responding to Reviewer 2/3's critique, which is to distinguish between sperm limitation and senescence and to confirm that there is both reproductive and body senescence. Would we have expected that both head-thorax and gaster to both senesce? We would have expected that head-thorax but not reproductive tissue to senesce. Therefore making the question explicit and some discussion of the predictions and findings would improve this section considerably.

We changed the Introduction, and explain now in the beginning of the Results section the purpose of the RNAseq analysis:

“To determine if queens show signs of reproductive senescence and loss of physiological function, we analyzed gene expression data of *prope mortem* queens exhibiting decreasing egg laying rates, and middle-aged queens that were at their peak reproductive performance.“

About the comparison between tissues, we included:

“We subjected the head plus thorax and the gaster (see methods for terminology, Figure 1 —figure supplement 5) to RNAseq separately, to asses if reproductive tissue shows different physiological wear and tear than head-thorax tissue”

Finally, how confident are we that the expression changes presented really indicate ageing. I would have appreciated more detailed justification. For example, you could include a table with the functions of key genes that are differentially expressed, and then we could see what down or up regulation in old queens would mean. Alternatively, you could choose a couple of examples in the text and expand upon them. For example, we understand that gene co-expression networks are associated with aging, but why? This is a key result that distinguishes thorax-head and gaster transcriptome and is central to your thesis later that thorax-head deteriorates physiologically while gaster maintains some functionality. Therefore, some more justification would allow the reader to better appreciate these changes. See also comment on Lines 247-249.

We considered adding such table of most important genes (besides the complete table now in Source data file 9 and 10), but realize that it is impossible to avoid cherry picking. This is the reason why we present a functional enrichment analysis, to give a clear perspective of the functions involved. Even so, the GOTerm results is quite extensive (see Supplement file 1B-E) and picking just a few functions to describe the changes with age does not seem very useful.

Discussion:Abstract and Line 246: Please remove the word "dramatic" before physiological changes, since you only have a 2-point and not 3-point comparison.

Done.

Lines 247-249: This statement is not well supported since they were not statistically different: "The changes are stronger in the gaster, which contains the reproductive organs and most of the digestive system, but to a similar extent occur in head and thorax, containing most neuronal and muscle tissue." Please refer explicitly to the evidence supporting this statement.

We have added that the tissues show differences in the number of duplicated reads (which does not seem to be a technical artifact). Nevertheless, we state that processes occurring in both tissues are similar in the Discussion as:

“Given these discrepancies between tissue types, 104 GO terms were significantly enriched in both tissues, of these 44 in *prope mortem* queens (Supplementary File 1F) and 60 in middle-aged queens (Supplementary File 1G). Thus, signs of similar physiological pathologies occur in reproductive and non-reproductive tissue.”

Line 251-252: Please provide some examples of these genes.

Done.

Conclusion:Lines 287-297: Once again, these predictions are placing singular focus on the queen, without taking workers into account, which in our view, is antithetical to the superorganism perspective. Without direct supporting evidence, you cannot rule out that the interaction between queens and workers is just as important to the superorganism perspective as is the queen herself. In our view, we ask you to maintain a neutral stance on this point and ask that you maintain the data on worker longevity that you removed in the previous version and revise the text.

We reintegrated the data on worker longevity to maintain the perspective on the superorganism, and agree there could be indirect interactions between queen and worker longevity.